# Occurrence and Molecular Characterization of *Cryptosporidium* spp. in Beef Cattle in Yunnan Province, China

**DOI:** 10.3390/microorganisms13040834

**Published:** 2025-04-07

**Authors:** Dongxu Li, Liujia Li, Jianfa Yang, Junjun He, Fengcai Zou, Fanfan Shu

**Affiliations:** 1The Yunnan Key Laboratory of Veterinary Etiological Biology, Yunnan Agricultural University, Kunming 650201, China; lidxu2000821@sina.com (D.L.); jsc315@163.com (J.Y.); hejunjun617@163.com (J.H.); 2Faculty of Animal Science and Technology, Yunnan Agricultural University, Kunming 650201, China; 3College of Agriculture and Biological Science, Dali University, Dali 671003, China; liliujia2007@163.com

**Keywords:** bovine, cryptosporidiosis, molecular epidemiology, risk factor, genotyping

## Abstract

*Cryptosporidium* spp. are protozoan pathogens that are widespread within mammals. In recent years, extensive molecular epidemiology studies on *Cryptosporidium* in dairy cattle have been conducted in Yunnan and worldwide. However, the infection status of these pathogens in beef cattle in Yunnan remains unclear. To examined the occurrence of *Cryptosporidium* spp. in beef cattle in Yunnan Province, China, we collected 735 fecal samples from six breeds of beef cattle in five regions of Yunnan. Nested PCR and DNA sequencing revealed the infection, species, and genotypes of *Cryptosporidium* spp. in these animals. The occurrence of *Cryptosporidium* spp. in Simmental cattle, Brahman cattle, Aberdeen Angus cattle, Yunnan Yellow cattle, Dulong cattle, and Hereford cattle was 32.9% (137/416), 3.8% (4/106), 24.4% (20/82), 3.8% (3/79), 3.2% (1/31), and 0% (0/21), respectively, with an overall rate of 22.4% (165/735). Regarding the regions, the occurrence of *Cryptosporidium* spp. in Boshan City, Kunming City, Lincang City, Dehong City and Xishuangbanna City was 41.8%, 28.6%, 19.4%, 6.7%, and 3.8%, respectively. In terms of age, the infection rates of *Cryptosporidium* spp. in pre-weaned, post-weaned, juvenile, and adult cattle were 62.1%, 52.6%, 42.7%, and 7.7%, respectively. According to sex, male cattle were more susceptible to *Cryptosporidium* infection (28.0%) than females (15.7%). Four *Cryptosporidium* species were identified in beef cattle: *C. andersoni* (*n* = 146), *C. bovis* (*n* = 11), *C. ryanae* (*n* = 7), and *C. occultus* (*n* = 1). Multilocus sequence typing analysis at the MS1, MS2, MS3, and MS16 gene loci revealed four subtype families of *C. andersoni* (A4A4A4A1, A5A4A4A1, A4A4A2A1, A1A4A4A1). Additionally, sequencing analysis of the 60-kDa glycoprotein gene identified three subtype families of *C. bovis* (XXVIc, XXVId, XXVIe) and one subtype family of *C. ryanae* (XXIb). These findings document the occurrence of *Cryptosporidium* spp. in beef cattle in Yunnan Province for the first time, providing reference data on the distribution, infection rate, species diversity, and genetic structure of these pathogens in China. To effectively reduce the prevalence of *Cryptosporidium* spp. in beef cattle in Yunnan, the implementation of proper sanitation management, rigorous rodent control, and farmer education programs is crucial. These integrated measures are critical for maintaining herd health, reducing economic losses, and ensuring meat safety across the province.

## 1. Introduction

*Cryptosporidium* spp. are important zoonotic protozoa that infect a wide range of vertebrates worldwide, including humans and domestic and wild animals [1,2]. *Cryptosporidium* oocysts are widely distributed in the environment, and can be ingested by susceptible humans and animals, resulting in cryptosporidiosis [3]. The main clinical feature of cryptosporidiosis is persistent diarrhea, and the disease has been reported in over 40 countries worldwide [4]. Bovine species are common hosts of these pathogens [5]. In beef cattle, cryptosporidiosis often results in high neonatal mortality rates and increased pharmaceutical costs, leading to significant economic losses in the livestock industry [6].

To date, approximately 44 valid species and 120 genotypes of *Cryptosporidium* spp. have been reported, with *C. andersoni*, *C. bovis*, *C. ryanae*, and *C. parvum* being the four major species found in cattle [7]. The prevalence of *Cryptosporidium* spp. in cattle varies globally, ranging from 6.25% to 39.6% [8]. Several factors influence the occurrence of *Cryptosporidium* spp. in cattle, including geographical location, climate, host age, herd management, food and water sources, sanitation, and rodent control [9].

Several subtyping tools for *Cryptosporidium* spp. play important roles in epidemiological studies [10,11]. The 60-kDa glycoprotein (*gp60*) gene-targeted subtyping method and multilocus sequence typing (MLST) have been reported as the most widely used tools for subtype identification of these pathogens [12,13]. Thus far, at least twenty, eight, and six different *gp60* subtypes have been identified in *C. parvum*, *C. ryanae*, and *C. bovis*, respectively [14,15,16]. In addition, more than 10 MLST subtypes of *C. andersoni* have been reported in cattle in China [17]. These diagnostic tools have been widely applied to study the transmission and diversity of *Cryptosporidium* spp. [18].

Yunnan Province, located in southwestern China, provides a high-quality production environment for beef cattle, due to its suitable geographical and climatic conditions [19]. As the largest beef cattle production province in China, it maintained over 9 million head of cattle in 2023, and is home to at least 15 different cattle breeds [19]. In previous studies, an indigenous breed of cattle (i.e., Yunling cattle) in Yunnan Province was found to be infected with *C. andersoni* and *C. ryanae* in a small-scale survey of animals [20]. However, the occurrence of these parasites in other beef cattle in Yunnan Province remains unknown. The “One Health” framework emphasizes integrated approaches to reduce disease risks at the human–animal–environment interface. A better understanding of *Cryptosporidium* species in beef cattle and their transmission potential can lead to better control strategies. These measures would improve animal health and production efficiency, thus supporting the growing demand for sustainable food production. Therefore, the aim of our study was to investigate the occurrence of *Cryptosporidium* spp. in six breeds of cattle from five regions in Yunnan, and to determine their species and subtypes in order to assess the transmission risks.

## 2. Materials and Methods

### 2.1. Sample Collection and Sampling Area

A total of 735 fecal samples from six breeds of beef cattle (i.e., Simmental cattle, Brahman cattle, Aberdeen Angus cattle, Yunnan Yellow cattle, Dulong cattle, Hereford cattle) from five regions in Yunnan Province, China, were used for sampling (Table 1 and Figure 1). Sampling in Baoshan, Dehong, and Xishuangbannna took place in June, while sampling in Lincang took place in September, and sampling in Kunming occurred in October 2024. All sampling periods coincided with the rainy season in Yunnan Province. Six representative breeds of beef cattle were selected for this study. Among them, Simmental and Aberdeen Angus are extensively raised in Yunnan, and serve as the main source of income for local farms. Yunnan Yellow, Hereford cattle, and Dulong cattle, known for their excellent meat quality, are primarily raised in small-scale farms in Yunnan. Additionally, Brahman cattle, imported from Laos due to their low cost, are temporarily raised in farms in Yunnan before being sold.

In the study regions, Simmental cattle and Aberdeen Angus cattle are managed in intensive farms with herd sizes of more than 1000 head. Yunnan Yellow, Hereford cattle, and Dulong cattle are managed in small-scale farms with herd sizes of less than 400 head. These animals are kept in confined places, and are fed a diet consisting of self-produced silage and commercially purchased concentrated feed. Brahman cattle are grazed in a special area of Xishuangbanna. In the Dehong and Kunming regions, cattle of two different breeds were sampled from two separate farms. The average annual rainfall in these five regions ranged from 720 mm to 1441 mm in 2024 (Table 1). Except for Xishuangbanna, which has a tropical climate, the other regions have a subtropical climate (Table 1).

Rectal sampling was performed separately for each beef cattle breed to avoid contamination. Each sample was placed in a separate self-sealing bag with clear information, such as the distribution, breed of cattle, date of birth, and gender. Collected samples were transported to a laboratory at 2 °C to 8 °C for no more than 72 h and stored in 2.5% potassium dichromate at 4 °C until DNA extraction.

### 2.2. DNA Extraction

Prior to the extraction of genomic DNA, 200 mg of fecal sample was placed in a centrifuge tube and centrifuged (2000× *g*, 10 min), washed with distilled water, and the supernatant was discarded to remove potassium dichromate. DNA was extracted from the pellet using a FastDNA Spin Kit for Soil (MP Biomedicals, Solon, OH, USA), as previously described [21]. The extracted DNA from the collected fecal samples was stored at −20 °C until analysis by PCR.

### 2.3. PCR Amplification

Nested PCR based on the small ribosomal subunit RNA (*SSU* rRNA) gene was used to detect and identify *Cryptosporidium* species in all DNA samples, with a target product size of approximately 830 bp [22]. Nested PCR was also employed for subtype identification of *C. bovis* and *C. ryanae* based on the *gp60* gene, with product sizes of approximately 1300 bp and 1000 bp [15,16]. *C. andersoni* was subtyped by analyzing four minisatellite/microsatellite targets (MS1, MS2, MS3, MS16) according to previous studies [17].

### 2.4. Sequence Analysis

Based on the gel electrophoresis results, all secondary positive PCR products were sent to Sangon Biotech (Kunming, China) for bidirectional sequencing on an ABI 3730 sequencer (Applied Biosystems, Foster City, CA, USA). Each raw sequence was assembled using ChromasPro 2.1.5.0 (http://technelysium.com.au/ChromasPro.html/, accessed on 23 October 2024). Assembled sequences were compared with GenBank dates to select appropriate reference sequences. ClustalX software 2.1.5.0 (http://clustal.org/, accessed on 23 October 2024) was used for comparative analysis between the sample sequences and the reference sequences. Detailed corrections of the sequences were performed using BioEdit 7.1 software (http://thalljiscience.github.io, accessed on 23 October 2024) to accurately determine the species and subtype of *Cryptosporidium* spp. in the samples [23]. A phylogenetic tree was constructed using MEGA 7.0 software (http://www.megasoftware.net/, accessed on 24 December 2024) based on the maximum likelihood method. The genetic relationship and reliability of the phylogenetic tree were assessed using the general time-reversible model and by bootstrapping with 1000 replicates, with values greater than 50% marked at the nodes [24]. The representative sequences were deposited in GenBank under accession numbers OL912798-OL912805, OM066896-OM066904, ON890790, and ON890791.

### 2.5. Statistical Analysis

The frequency of *Cryptosporidium* spp. occurrence among regions, breeds, ages, and genders were calculated using chi-squared tests implemented in SPSS20.0 (IMB SPSS Int, Chicago, IL, USA) and SAS9.1 (SAS Institute Inc., Cary, NC, USA). Statistical difference was considered significant at *p* < 0.05. Odds ratios (ORs) with 95% confidence intervals (CIs) were calculated to assess whether the associated factors were risk factors.

## 3. Results

### 3.1. Occurrence of Cryptosporidium Species

*Cryptosporidium* spp. were found in 165 (22.4%) of the 735 beef cattle fecal samples from five cities in Yunnan Province (Table 2). Among them, the infection rate in Baoshan (41.8%) was significantly higher than in Kunming (28.6%, *χ*^2^ = 6.787, *p* = 0.01114), Lincang (19.4%, *χ*^2^ = 15.146, *p* = 0.00011), Dehong (6.7%, *χ*^2^ = 57.73, *p* < 0.00001), and Xishuangbanna (3.8%, *χ*^2^ = 49.054, *p* < 0.00001). The occurrence in the six beef cattle breeds ranged from 0% to 32.9%. The detection rate in Simmental cattle (32.9%) was significantly higher than in Aderdeen Angus cattle (24.4%, *χ*^2^ = 2.315, *p* = 0.1526), Yunnan Yellow cattle (3.8%, *χ*^2^ = 27.784, *p* < 0.00001), Brahman cattle (3.8%, *χ^2^* = 36.431, *p* < 0.00001), Dulong cattle (3.2%, *χ*^2^ =11.930, *p* = 0.00017), and Hereford cattle (0%).

The occurrence in pre-weaned calves (62.1%, *χ*^2^ = 89.090, *p* < 0.00001), post-weaned calves (52.6%, *χ*^2^ = 92.789, *p* < 0.00001), and juvenile cattle (42.7%, *χ*^2^ = 112.391, *p* < 0.00001) was significantly higher than in adults (7.7%). In terms of gender, the infection rate in male cattle (28.0%, *χ*^2^ =15.709, *p* = 0.0008) was significantly higher than in female cattle (15.7%). Thus, all the studied factors (i.e., location, breed, age and sex) significantly influenced the occurrence of *Cryptosporidium* spp. in beef cattle in Yunnan.

### 3.2. Identification of Cryptosporidium Species

Small subunit rRNA (*SSU* rRNA) sequence analysis was conducted on *Cryptosporidium*-positive specimens. As shown in Table 3, *C. andersoni* was identified in 19.8%, (146/735) of the samples, while *C. bovis* accounted for 1.4% (11/735), *C. ryanae* for 0.9% (7/735), and *C. occultus* for 0.1% (1/735) in beef cattle fecal samples collected from the five regions in Yunnan Province. *C. andersoni* was detected in all areas; *C. bovis* and *C. ryanae* were found in Baoshan, Dehong, and Kunming; and *C. occultus* was exclusively identified in Baoshan. By breed, *C. andersoni* was the most prevalent species in four beef cattle breeds, whereas *C. ryanae* was found only in Simmental cattle and Yunnan yellow cattle. In contrast, *C. bovis* and *C. occultus* were exclusively found in Simmental cattle. Meanwhile, both *C. andersoni* and *C. bovis* were detected across all age groups, whereas *C. ryanae* was absent in juveniles, and *C. occultus* was found only in juveniles. Additionally, *C. andersoni*, *C. bovis*, and *C. ryanae* were detected in both male and female cattle, while *C. occultus* was exclusively found in male cattle. The nucleotide sequences obtained from *C. andersoni*, *C. bovis*, *C. ryanae*, and *C. occultus* were identical to GenBank sequences JN400881, MT950118, JN400880, and MK982467, respectively. As expected, these *Cryptosporidium* spp. sequences were placed in relation to their reference sequence in the phylogenetic analysis of the *SSU* rRNA gene (Figure 2).

### 3.3. Subtyping of Cryptosporidium *spp*.

A total of 146 *C. andersoni* positive samples were subtyped using MLST at four loci (i.e., MS1, MS2, MS3, and MS16). From these, 45 *C. andersoni* specimens were successfully classified into four distinct MLST subtypes. The most prevalent subtype identified was A4A4A4A1 (*n* = 23), which was predominantly found in Simmental cattle and Aderdeen Angus cattle. This was followed by the subtypes A5A4A4A1 (*n* = 11) and A4A4A2A1 (*n* = 9), and A1A4A4A2 (*n* = 2), in Simmental cattle and Brahman cattle (Table 2). The subtype A1A4A4A2 (*n* = 2) was detected in female cattle, while the other three subtypes were found in both male and female cattle. Moreover, samples positive for *C. bovis* and *C. ryanae* were analyzed using the *gp60* locus for subtyping. Of the eleven *C. bovis*-positive samples collected in Simmental cattle from Kunming and Baoshan, six were successfully categorized into three genetic groups: XXVIc (*n* = 4), XXVId (*n* = 1), and XXVIe (*n* = 1). For *C. ryanae*, four out of seven positive samples were identified, with only one subtype, XXIb, detected in Simmental cattle from Kunming (Table 3).

## 4. Discussion

Our results suggest that *Cryptosporidium* spp. are prevalent in beef cattle in Yunnan Province. The occurrence of *Cryptosporidium* spp. in this study (22.4%) was higher than that reported in Bangladesh (5.0%) [25] and Egypt (10.2%) [26], but lower than that reported in Japan (77.5%) [27], Scotland (59.3%) [28], Italy (38.8%) [29], and Estonia (23.0%) [30], as well as being lower than the global prevalence of bovine cryptosporidiosis (29.1%) [31]. Within China, our findings were lower than those in Taiwan (37.6%) [32] and Henan (26.5%) [33], but higher than those in Shaanxi (20.2%) [34], Heilongjiang (17.5%) [35], Hubei (15.6%) [36], Jiangxi (12.8%) [37], Shanxi (11.1%) [38], and the two reports of pooled prevalence in China (14.5% and 8.0%) [24,39]. *Cryptosporidium* spp. infection rates demonstrated regional variability, influenced by factors such as geographical location, herd size, animal age, sampling time, total sample size, study design, climate, and sanitation conditions [40,41].

The occurrence of *Cryptosporidium* spp. in calves aged 0–18 months was significantly higher than that in adult cattle, with the highest positive rate observed in pre-weaned calves. This suggests that younger animals are more prone to infection, which is consistent with previous studies [42,43]. Furthermore, a significantly higher incidence of *Cryptosporidium* spp. was detected in Simmental cattle and Aberdeen Angus cattle than in other local beef breeds. This might be due to the fact that these two common beef breeds were reared in intensive farming systems, whereas other local beef breeds were reared mainly on small-scale farms or under free-range conditions. Several previous reviews have reported that concentrated animal feeding operations have been shown to be conducive to the transmission of pathogens, due to large numbers of susceptible animals in confined spaces, together with low genetic diversity of animals [44,45]. In recent years, Simmental cattle have been the dominant beef cattle breed and widely distributed in Yunnan Province [46]. This study collected a larger number of samples from this breed in three regions, which may be a contributing factor to the observed higher occurrence. In further studies, we need to balance sample sizes across breeds to better understand the role of breed-specific factors in *Cryptosporidium* spp. infection. Most studies have shown that the occurrence of *Cryptosporidium* spp. is highly sensitive to climatic conditions, such as temperature, rainfall, and humidity [47]. However, we did not observe a similar phenomenon in this study. In Xishuangbanna, a region with a tropical climate characterized by higher temperature and humidity, the occurrence of *Cryptosporidium* spp. in cattle was the lowest. This may be because the factors influencing infection rates are multifaceted.

Four *Cryptosporidium* species were identified, including *C. andersoni*, *C. bovis*, *C. ryanae*, and *C. occultus*. Among these, *C. andersoni* was the dominant species in beef cattle, which is consistent with previous findings from many regions of China, India, and Spain [48,49,50]. In contrast, *C. bovis* has been identified as the predominant species in beef cattle in other countries [51,52,53]. Compared to calves, *C. andersoni* was more prevalent in adult cattle [54]. Infections caused by this species could lead to gastritis, poor weight gain, and adverse effects on production efficiency [55]. However, the present study showed that *C. andersoni* was the dominant species in all age groups of beef cattle, a phenomenon that may be attributed to the mixed housing of adult cattle with calves. In addition, *C. occultus* has been reported in humans, dairy cattle, water buffalo, yaks, and rats [56,57,58,59,60]. Among these, rodents might serve as the primary host for *C. occultus* [61]. Some studies have found that the presence of *C. occultus* in cattle is attributed to their consumption of food contaminated with rodent feces [62]. Therefore, despite the low detection rate of *C. occultus* in this study, it is indicative of inadequate rodent control measures on farms. Although *C. parvum* was one of the top four species infecting cattle worldwide, no cases of *C. parvum* were detected in our study. Previous research has suggested that *C. parvum* is more prevalent in certain dairy farms, because of the relatively short history of intensive livestock production in China [63].

A common diversity of *Cryptosporidium* spp. was detected in this study, with four MLST subtypes of *C. andersoni* identified. One MLST subtype (A4,A4,A4,A1) was the dominant subtype in beef cattle, which was also prevalent in dairy cattle in this province [64]. This confirms previous findings that the (A4,A4,A4,A1) subtype was the most prevalent in China, including in the Guangdong [65], Heilongjiang [36], and Shaanxi Provinces [66]. The three MLST subtypes (A5A4A4A1, A4A4A2A1, and A1A4A4A1) detected in the present study have also been reported in dairy cattle from Xinjiang Province [67] and Yunnan Province [68]. The present study also reported three *C. bovis* subtype families (XXVId, XXVIc, and XXVIe) that are similar to those reported in Henan Province [16]. This finding corroborates other observations of a lack of geographical segregation and host adaptation in *C. bovis* [16]. The *C. bovis* subtype XXVIc was prevalent in both pre-weaned and post-weaned calves in the present study, aligning with previous longitudinal studies conducted in Guangzhou [68]. This indicates that the distribution of *C. bovis* subtypes is influenced by host age and subtype-specific immunity [43]. To date, only two subtypes of *C. ryanae* (XXIa, XXIf) have been reported in beef cattle in Shijiazhuang [15], with XXIa identified for the first time in this study. Compared to dairy cattle in Yunnan, a low diversity of *C. bovis* and *C. ryanae* has been detected in beef cattle in this province [69].

To meet the growing demand for meat products, Yunnan Province has significantly expanded the beef cattle industry. However, large numbers of animals in confined spaces could promote the transmission of *Cryptosporidium* spp. Both farmers and consumers should be aware of cattle-associated *Cryptosporidium* species, as these pathogens can spread through fecal contamination of water, soil, and food, potentially exposing humans to infectious oocysts. This poses a particular risk to immunocompromised individuals, as there is no effective treatment against cryptosporidiosis. Our study identified key risk factors, including overcrowding, poor sanitation, and insufficient rodent control, all of which contribute to the transmission of *Cryptosporidium* spp. in beef cattle. Only through implementing the “One Health” approach, which emphasizes human–animal–environment interactions, can *Cryptosporidium* infections be effectively mitigated [70].

## 5. Conclusions

The results of this study reveal a higher prevalence of *Cryptosporidium* spp. in Yunnan beef cattle, with *C. andersoni* as the dominant species. Infection rates were significantly higher in Simmental cattle, pre-weaned calves, and male cattle, highlighting breed, age, and sex as key risk factors. The four *C. andersoni* subtypes (A4,A4,A4,A1, A5,A4,A4,A1, A4,A4,A2,A1, and A1,A4,A4,A1), three *C. bovis* subtypes (XXVId, XXVIc, and XXVIe) and one *C. ryanae* subtype (XXIa) were identified in this province. The precise differentiation of these subtypes facilitates a more profound comprehension of the transmission mechanisms and pathogenic characteristics of pathogens. These results improve our understanding of the occurrence of *Cryptosporidium* spp. in beef cattle in Yunnan, providing a fundamental dataset for the control of the transmission of these pathogens in the province.

## Figures and Tables

**Figure 1 microorganisms-13-00834-f001:**
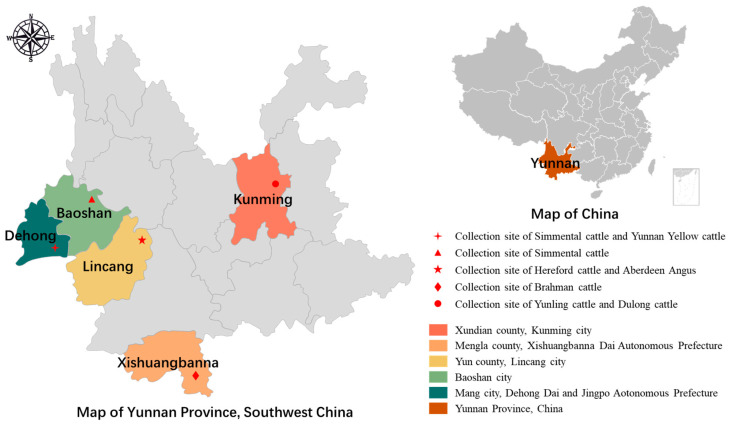
Map of beef cattle sampling sites in Yunnan Province, China.

**Figure 2 microorganisms-13-00834-f002:**
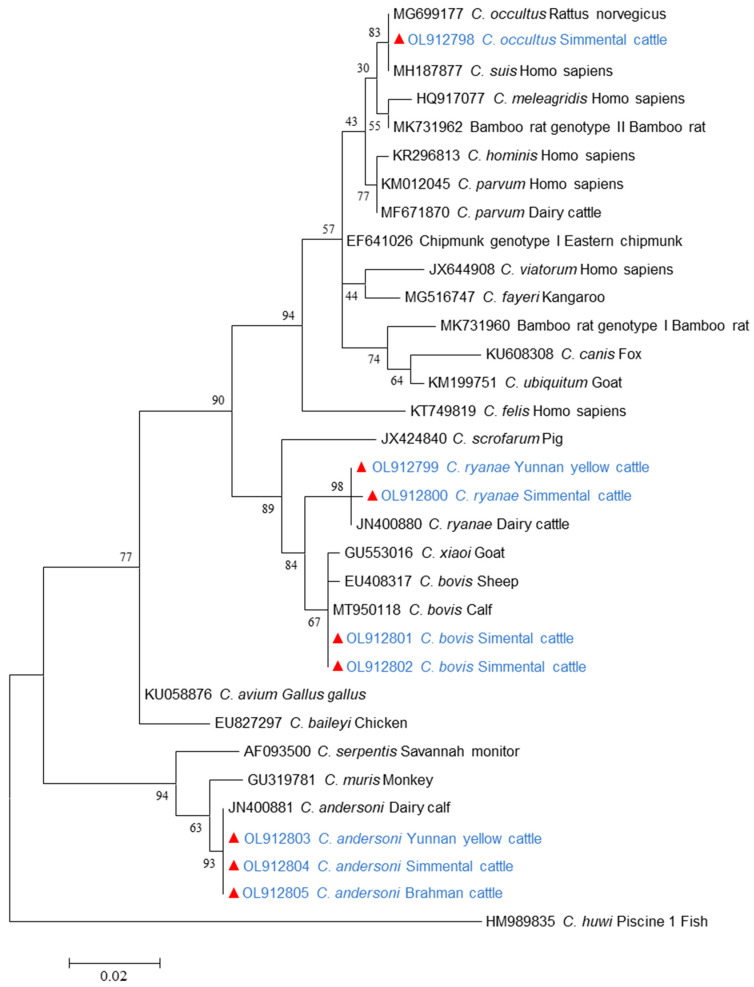
Maximum-likelihood phylogenetic tree of *Cryptosporidium* species based on 18S rRNA gene sequence. Sequences identified in this study are marked by filled red symbols.

**Table 1 microorganisms-13-00834-t001:** Geographical distribution and sample collection details of six species of beef cattle at five locations in Yunnan Province, China, for *Cryptosporidium* spp. investigations.

Location	Geographical Coordinates	Altitude (m)	Climate	Rainfall in 2024 (mm)	No. of Samples						Total
					Simmental Cattle	Brahman Cattle	Aberdeen Angus Cattle	Yunnan Yellow Cattle	Dulong Cattle	Hereford Cattle	
Baoshan	98°98′ E, 25°18′ N	730–1800	Subtropical	1060	201	-	-	-	-	-	201
Dehong	98°48′ E, 24°33′ N	835–2800	Subtropical	1110	85	-	-	79	-	-	164
Kunming	102°9′ E, 25°63′ N	2100–2900	Subtropical	720	130	-	-	-	31	-	161
Xishuangbanna	101°55′ E, 21°52′ N	640–1200	Tropical	1441	-	106	-	-	-	-	106
Lincang	99°95′ E, 24°14′ N	740–1400	Subtropical	1037	-	-	82	-	-	21	103
Total	-	-			416	106	82	79	31	21	735

**Table 2 microorganisms-13-00834-t002:** Occurrence and factors associated with *Cryptosporidium* spp. infection in beef cattle in Yunnan Province, China.

Variable	Category	No. Tested	No. Positive	Prevalence (%) (95% CI)	OR (95%, CI)	*p*-Value
Region	Baoshan	201	84	41.8 (34.91–48.67)	18.3 (6.49–51.67)	<0.01
	Dehong	164	11	6.7 (2.84–10.58)	1.83 (0.57–5.92)	
	Kunming	161	46	28.6 (21.52–35.62)	10.2 (3.55–29.32)	
	Xishuangbanna	106	4	3.8 (0.09–7.46)	Reference	
	Lincang	103	20	19.4 (11.65–27.19)	6.1 (2.02–18.68)	
Breed	Simmental cattle	416	137	32.9 (28.40–37.45)	14.7 (1.99–109.16)	<0.01
	Brahman cattle	106	4	3.8 (0.09–7.46)	1.2 (0.13–10.93)	
	Aberdeen Angus cattle	82	20	24.4 (14.90–33.88)	9.7 (1.24–75.56)	
	Yunnan Yellow cattle	79	3	3.8 (−0.51–8.11)	1.2 (0.12–11.84)	
	Dulong cattle	31	1	3.2 (−3.36–9.81)	Reference	
	Hereford cattle	21	-	-	-	
Age	Pre-weaned (0–60 days)	29	18	62.1(43.29–80.85)	19.7 (8.64–45.04)	<0.01
	Post-weaned (61–180 days)	57	30	52.6 (39.27–66.00)	13.4 (7.18–25.00)	
	Juvenile cattle (7–18 months)	192	82	42.7 (35.64–49.77)	9.0 (5.74–14.07)	
	Adult cattle (>18 months)	457	35	7.7 (5.21–10.11)	Reference	
Sex	Male	404	113	28.0 (23.57–32.37)	2.1 (1.44–3.01)	<0.01
	Female	331	52	15.7 (11.77–19.65)	Reference	
Total		735	165	22.4 (19.42–25.47)	-	-

No: number; CI: confidence interval; OR: odds ratio.

**Table 3 microorganisms-13-00834-t003:** Species and subtype identification of *Cryptosporidium* spp. from beef cattle in Yunnan Province, China.

FACTORS	CATEGORY	*CRYPTOSPORIDIUM* SPECIES	*CRYPTOSPORIDIUM* SUBTYPES		
			*C. andersoni*	*C. bovis*	*C. ryanae*
REGION	Baoshan	*C. andersoni* (76), *C. bovis* (6), *C. ryanae* (1), *C. occulutus* (1)	A4A4A4A1 (16)	XXVId (1), XXVIe (1)	-
	Dehong	*C. andersoni* (8), *C. ryanae* (2), *C. bovis* (1)	A1A4A4A1 (2)	-	-
	Kunming	*C. andersoni* (38), *C. bovis* (4), *C. ryanae* (4)	A5A4A4A1 (11), A4A4A2A1 (8), A4A4A4A1 (4)	XXVIc (4)	XXIb (4)
	Xishuangbanna	*C. andersoni* (4)	A4A4A2A1 (1)	-	-
	Lincang	*C. andersoni* (20)	A4A4A4A1 (3)	-	-
BREED	Simmental cattle	*C.andersoni* (119), *C. bovis* (11), *C. ryanae* (6), *C. occulutus* (1)	A4A4A4A1 (20), A5A4A4A1 (11), A4A4A2A1 (8), A1A4A4A1 (2)	XXVIc (4), XXVId (1), XXVIe (1)	XXIb (4)
	Brahman cattle	*C. andersoni* (4)	A4A4A2A1 (1)	-	-
	Aberdeen Angus cattle	*C. andersoni* (20)	A4A4A4A1 (3)	-	-
	Yunnan Yellow cattle	*C. andersoni* (2), *C. ryanae* (1)	-	-	-
	Dulong cattle	*C. andersoni* (1)	-	-	-
	Hereford cattle	-	-	-	-
AGE	Pre-weaned (0–60 days)	*C. andersoni* (12), *C. bovis* (3), *C. ryanae* (3)	A4A4A2A1 (3), A5A4A4A1 (2)	XXVIc (3)	XXIb (3)
	Post-weaned (61–180 days)	*C. andersoni* (27), *C. ryanae* (2), *C. bovis* (1)	A5A4A4A1 (8), A4A4A4A1 (6), A4A4A2A1 (2)	XXVIc (1)	XXIb (1)
	Juvenile cattle (7–18 months)	*C. andersoni* (78), *C. bovis* (3), *C.occulutus* (1)	A4A4A4A1 (16), A5A4A4A1 (1)	XXVId (1), XXVIe (1)	-
	Adult cattle (>18 months)	*C. andersoni* (29), *C. bovis* (4), *C. ryanae* (2)	A4A4A2A1 (4), A1A4A4A1 (2), A4A4A4A1 (1)	-	-
SEX	Male	*C. andersoni* (104), *C. bovis* (5), *C. ryanae* (3), *C.occulutus* (1)	A4A4A4A1 (20), A5A4A4A1 (6), A4A4A2A1 (2)	XXVIc (2), XXVId (1), XXVIe (1)	XXIb (2)
	Female	*C. andersoni* (42), *C. bovis* (6), *C. ryanae* (4)	A4A4A2A1 (7), A5A4A4A1 (5), A4A4A4A1 (3), A1A4A4A1 (2)	XXVIc (2)	XXIb (2)
TOTAL		*C.andersoni* (146), *C. bovis* (11), *C. ryanae* (7), *C. occulutus* (1)	A4A4A4A1 (23), A5A4A4A1 (11), A4A4A2A1 (9), A1A4A4A1 (2)	XXVIc (4), XXVId (1), XXVIe (1)	XXIb (4)

## Data Availability

The datasets presented in this study can be found in online repositories.

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
