# Peer review of "Occurrence and Molecular Characterization of Cryptosporidium spp. in Beef Cattle in Yunnan Province, China"

_microorganisms, 2025, doi:10.3390/microorganisms13040834_

Round 1

Reviewer 1 Report

Comments and Suggestions for Authors

Specify, in the Introduction, which are the favourable factors for this infection in order to justify the correlations made in this study.

WARNING! At the beginning of this study, in the Introduction, you wrote that you have chosen 7 breeds, and in Materials and Methods you talk about 6 breeds. Explain why you have chosen 6 breeds of beef cattle out of the 15 breeds that you say exist in Yunnan Province.

It is necessary to present, at Materials and Methods, the animal husbandry conditions in each region under study. It would also be important to mention how the animals are foraged, where the animals get their feed and whether they are left to graze. Open access of animals may influence the occurrence of infection.

Also, at Materials and methods you should present in a few words the geographical, climatic and rainfall conditions in each region under study. And then, at Discussion, mention whether temperature or humidity in each region may influence the prevalence of infection.

It

You need to present more detailed correlations between the prevalence of infection and the age of the animal, gender or region of the province.

You have not mentioned anything about prevalence of infection by animal's gender.

Please present the authors' opinion on the correlation between the prevalence of infection and the breed of the animal in the Discussion. You wrote that the Simmental breed showed the highest prevalence. But the number of cattle in the study of this breed is significantly higher (no. 416) compared to the other breeds. Could this also be a factor influencing the rate of positive samples identified in the Simmental breed?

For the regions of Dehon and Kunming, where you have cattle of two different breeds in your study, you should specify whether the animals came from the same farm or from different farms, specify the maintenance conditions for each breed and give the number of positive samples in each of the two regions by breed.

Both in the Abstract and in the Conclusions, the authors should present the prevalence of prevalence by age and gender of the animal and by region, not only by breed.

would be important to mention whether the samples were all collected in the same season or at different times of the calendar year. Even this may influence the prevalence.

Author Response

Response to Reviewer 1 Comments

1. Summary

Thank you very much for taking the time to review this manuscript. Please find the detailed responses below and the corresponding revisions in the re-submitted files.

2. Questions for General Evaluation

Reviewer’s Evaluation

Response and Revisions

Does the introduction provide sufficient background and include all relevant references?

Must be improved

Improved in Page 1, Lines 55-58

Are all the cited references relevant to the research?

Yes

Many thanks

Is the research design appropriate?

Can be improved

Many thanks

Are the methods adequately described?

Must be improved

Improved in Page 2, Lines 87-104

Are the results clearly presented?

Can be improved

Improved in Page 5, Lines 156-169

Are the conclusions supported by the results?

Can be improved

Improved in Page 1, Lines 37-41

3. Point-by-point response to Comments and Suggestions for Authors

Comments 1: [Specify, in the Introduction, which are the favourable factors for this infection in order to justify the correlations made in this study. ]

Response 1: Thank you for your constructive feedback. [The prevalence of Cryptosporidium spp. in cattle varies globally, ranges from 6.25% to 39.6% [8]. Several factors influence the occurrence of Cryptosporidium spp. in cattle, including geographical location, climate, host age, herd management, food and water sources, sanitation, and rodent control [9].] We have added the information in the introduction in Page 2, Paragraph 2, Lines 55-58.

Comments 2: [WARNING! At the beginning of this study, in the Introduction, you wrote that you have chosen 7 breeds, and in Materials and Methods you talk about 6 breeds. Explain why you have chosen 6 breeds of beef cattle out of the 15 breeds that you say exist in Yunnan Province. ]

Response 2: Thanks you very much for pointing out our oversight. [Initially, the study aimed to investigate seven beef cattle breeds, including Yunling cattle. However, during the drafting of the manuscript, we found that Yunling cattle had already been surveyed in a 2021 study (Microb Pathog. 2021 Sep;158:105025.), and the results were similar to ours. Therefore, we decided to exclude the data related to Yunling cattle from our manuscript to avoid redundancy. Unfortunately, we forget to remove the mention of Yunling cattle from the introduction.] We have corrected this in the introduction in Page 2, Paragraph 4, Lines 80.

  [Six representative breeds of beef cattle were selected for this study. Among them, Simmental, Aberdeen Angus were extensively raised in Yunnan and serve as the main source of income for local farms. Yunnan Yellow, Hereford cattle, and Dulong cattle, known for their excellent meat quality, were primarily raised in small-scaled farms in Yunnan. Additionally, Brahman cattle, imported from Laos due to their low cost, are temporarily raised in farms in Yunnan before being sold.] We have added the information in the material and method in Page 2-3, Paragraph 1, Lines 90-95.

Comments 3: [It is necessary to present, at Materials and Methods, the animal husbandry conditions in each region under study. It would also be important to mention how the animals are foraged, where the animals get their feed and whether they are left to graze. Open access of animals may influence the occurrence of infection. ]

Response 3: Thank you for pointing this out. [In the study regions, Simmental cattle and Aberdeen Angus cattle were managed in intensive farms with herd sizes of more than 1000 heads. Yunnan Yellow, Hereford cattle, and Dulong cattle are managed in small-scaled farms with herd sizes of less than 400 heads. These animals are kept in confined places and fed a diet consisting of self-produced silage and commercially purchased concentrated feed. Brahman cattle were grazed in a special area of Xishuangbanna.] We have added the information in Page 3, Paragraph 2, Lines 96-101.

Comments 4: [Also, at Materials and methods you should present in a few words the geographical, climatic and rainfall conditions in each region under study. And then, at Discussion, mention whether temperature or humidity in each region may influence the prevalence of infection. ]

Response 4: Thank you for pointing this out. [The average annual rainfall in these five regions ranged from 720 mm to 1441 mm in 2024 (Table 1). Except for Xishuangbanna, which has a tropical climate, the other regions have a subtropical climate (Table 1).] We have added the information in materials and methods in Page 3, Paragraph 2, Lines 102-104.

[Most studies have shown that the occurrence of Cryptosporidium spp. is highly sensitive to climatic conditions, such as temperature, rainfall and humidity [47]. However, we did not observe a similar phenomenon in this study. In Xishuangbanna, a region with a tropical climate characterized by higher temperature and humidity, the occurrence of Cryptosporidium spp. in cattle was the lowest. This may be because the factors influencing infection rates are multifaceted.] We have added the information in discussion in Page 9, Paragraph 3, Lines 239-244.

Comments 5: [You need to present more detailed correlations between the prevalence of infection and the age of the animal, gender or region of the province. ]

Response 5: Thank you for pointing this out. [Among them. the infection rate in Baoshan (41.8%) was significantly higher than in Kunming (28.6%, c2 = 6.787, P = 0.01114), Lincang (19.4%, c2 = 15.146, P = 0.00011), Dehong (6.7%, c2 = 57.73, P < 0.00001), and Xishuangbanna (3.8%, c2 = 49.054, P < 0.00001). The occurrence of six beef cattle breeds ranges from 0% to 32.9%. The detection rate in Simmental cattle (32.9%) was significantly higher than in Aderdeen Angus cattle (24.4%, c2 = 2.315, P = 0.1526), Yunnan Yellow cattle (3.8%, c2 = 27.784, P < 0.00001), Brahman (3.8%, c2 = 36.431, P < 0.00001), Dulong cattle (3.2%, c2 =11.930, P = 0.00017), and Hereford cattle (0%).

The occurrence in pre-weaned calves (62.1%, c2 = 89.090, P < 0.00001), post-weaned calves (52.6%, c2 = 92.789, P < 0.00001), juvenile cattle (42.7%, c2 = 112.391, P < 0.00001) was significantly higher than in adults (7.7%). By gender, the infection rate in male cattle (28.0%, c2 =15.709, P = 0.0008) was significantly higher than in female cattle (15.7%). Thus, all the studied factors (i.e., location, breed, age and sex) significantly influenced the occurrence of Cryptosporidium spp. in beef cattle in Yunnan.] We have added the information in results in Page 5, Paragraph 1-2, Lines 156-169.

Comments 6: [You have not mentioned anything about prevalence of infection by animal's gender. ]

Response 6: Thank you for pointing this out. We have added the information in results in Table 2 (Page 6, Lines 202) and Table 3 (Page 6, Lines 205).

Comments 7: [Please present the authors' opinion on the correlation between the prevalence of infection and the breed of the animal in the Discussion. You wrote that the Simmental breed showed the highest prevalence. But the number of cattle in the study of this breed is significantly higher (no. 416) compared to the other breeds. Could this also be a factor influencing the rate of positive samples identified in the Simmental breed? ]

Response 7: Agree. [In recent years, Simmental cattle have been dominating beef cattle breed and were widely distributed in Yunnan Province [46]. This study collected a larger number of samples from this breed in three regions, which may be a contributing factor to the observed higher occurrence. In further studies, we need to balance sample sizes across breeds to better understand the role of breed-specific factors in Cryptosporidium spp. infection.] We have added the information in Discussion in Page 9, Paragraph 2, Lines 233-238.

Comments 8: [For the regions of Dehon and Kunming, where you have cattle of two different breeds in your study, you should specify whether the animals came from the same farm or from different farms, specify the maintenance conditions for each breed and give the number of positive samples in each of the two regions by breed. ]

Response 8: Thank you for pointing this out. [In the Dehong and Kunming regions, cattle of two different breeds were sampled from two separate farms.] We have added the information in materials and methods in Page 3, Paragraph 2, Lines 101-102.

The maintenance conditions were added in Page 3, Paragraph 2, Lines 96-101.

Positive samples were present in table 2 (Page 6, Lines 202). Notably, Yunnan yellow cattle were only sampled in Dehong, while Dulong cattle were only sampled in Kunming. Thus, the data for Yunnan yellow cattle and Dulong cattle in Table 2 are region-specific.

Comments 9: [Both in the Abstract and in the Conclusions, the authors should present the prevalence of prevalence by age and gender of the animal and by region, not only by breed. ]

Response 9: Thank you for pointing this out. [Regarding the region, the occurrence of Cryptosporidium spp. in Boshan City, Kunming City, Lincang City, and Xishuangbanna City was 41.8%, 28.6%, 19.4%, 6.7% and 3.8%, respectively. In terms of age, the infection rates of Cryptosporidium spp. in pre-weaned, post-weaned, juvenile, and adult cattle were 62.1%, 52.6%, 42.7% and 7.7%, respectively. According to sex, male cattle were more susceptible to Cryptosporidium infection (28.0%) than females (15.7%).] We have added the information in abstract in Page 10, Lines 299-302.

[The results of this study reveal a higher prevalence of Cryptosporidium spp. in Yunnan beef cattle, with C. andersoni as the dominant species. Infection rates were significantly higher in Simmental cattle, pre-weaned calves and male cattle, highlighting breed, age, and sex as key risk factors.] We have added the information in conclusion in Page 1, Lines 24-29.

Comments 10: [Would be important to mention whether the samples were all collected in the same season or at different times of the calendar year. Even this may influence the prevalence. ]

Response 10: Thank you for pointing this out. [Sampling in Baoshan, Dehong, and Xishuangbannna took place in June, while sampling in Lincang took place in September, and sampling in Kunming occurred in October 2024. All sampling periods coincided with the rainy season in Yunnan Province.] We have added the information in method in Page 2, Paragraph 1, Lines 87-89.

4. Response to Comments on the Quality of English Language

Point 1: The English is fine and does not require any improvement.

Response 1: There is no content that requires a reply.

5. Additional clarifications

[There is no content that requires additional clarifications.]

Reviewer 2 Report

Comments and Suggestions for Authors

The bibliography is adequate. The study evaluates the presence of Cryptosporidium spp. in beef cattle in Yunnan Province, China. The introduction provides a detailed description of cryptosporidiosis in cattle, the potential risks for the livestock industry, and the importance of subtyping to study the transmission and diversity of Cryptosporidium spp. The number of significant samples, the sampling method is well described, as well as the laboratory techniques.The results are presented clearly and the conclusions are consistent with the results. I would like, if possible, to have some insights on potential environmental or management factors that could influence the presence of Cryptosporidium in beef cattle in the studied region and on differences based on age and breed.

Author Response

Response to Reviewer 2 Comments

1. Summary

Thank you very much for taking the time to review this manuscript. Please find the detailed responses below and the corresponding revisions in the re-submitted files.

2. Questions for General Evaluation

Reviewer’s Evaluation

Response and Revisions

Does the introduction provide sufficient background and include all relevant references?

Yes

Many thanks

Is the research design appropriate?

Yes

Many thanks

Are the methods adequately described?

Yes

Many thanks

Are the results clearly presented?

Can be improved

Improved in Page 2-3/9, Lines 56-58/ 87-104/233-244

Are the conclusions supported by the results?

Yes

Many thanks

3. Point-by-point response to Comments and Suggestions for Authors

Comments 1: [The bibliography is adequate. The study evaluates the presence of Cryptosporidium spp. in beef cattle in Yunnan Province, China. The introduction provides a detailed description of cryptosporidiosis in cattle, the potential risks for the livestock industry, and the importance of subtyping to study the transmission and diversity of Cryptosporidium spp. The number of significant samples, the sampling method is well described, as well as the laboratory techniques. The results are presented clearly and the conclusions are consistent with the results. I would like, if possible, to have some insights on potential environmental or management factors that could influence the presence of Cryptosporidium in beef cattle in the studied region and on differences based on age and breed. ]

Response 1: Thank you for your constructive feedback and recognition.

[Sampling in Baoshan, Dehong, and Xishuangbannna took place in June, while sampling in Lincang took place in September, and sampling in Kunming occurred in October 2024. All sampling periods coincided with the rainy season in Yunnan Province. Six representative breeds of beef cattle were selected for this study. Among them, Simmental, Aberdeen Angus were extensively raised in Yunnan and serve as the main source of income for local farms. Yunnan Yellow, Hereford cattle, and Dulong cattle, known for their excellent meat quality, were primarily raised in small-scaled farms in Yunnan. Additionally, Brahman cattle, imported from Laos due to their low cost, are temporarily raised in farms in Yunnan before being sold.

In the study regions, Simmental cattle and Aberdeen Angus cattle were managed in intensive farms with herd sizes of more than 1000 heads. Yunnan Yellow, Hereford cattle, and Dulong cattle are managed in small-scaled farms with herd sizes of less than 400 heads. These animals are kept in confined places and fed a diet consisting of self-produced silage and commercially purchased concentrated feed. Brahman cattle were grazed in a special area of Xishuangbanna. In the Dehong and Kunming regions, cattle of two different breeds were sampled from two separate farms. The average annual rainfall in these five regions ranged from 720 mm to 1441 mm in 2024 (Table 1). Except for Xishuangbanna, which has a tropical climate, the other regions have a subtropical climate (Table 1).] We have added the information in the Materials and Methods in Page 2-3, Paragraph 1, Lines 87-104.

[In recent years, Simmental cattle have been dominating beef cattle breed and were widely distributed in Yunnan Province [46]. This study collected a larger number of samples from this breed in three regions, which may be a contributing factor to the observed higher occurrence. In further studies, we need to balance sample sizes across breeds to better understand the role of breed-specific factors in Cryptosporidium spp. infection.

Most studies have shown that the occurrence of Cryptosporidium spp. is highly sensitive to climatic conditions, such as temperature, rainfall and humidity [47]. However, we did not observe a similar phenomenon in this study. In Xishuangbanna, a region with a tropical climate characterized by higher temperature and humidity, the occurrence of Cryptosporidium spp. in cattle was the lowest. This may be because the factors influencing infection rates are multifaceted.] We have added the information in the Discussion in Page 9, Paragraph 2-3, Lines 233-244.

[The results of this study reveal a higher prevalence of Cryptosporidium spp. in Yunnan beef cattle, with C. andersoni as the dominant species. Infection rates were significantly higher in Simmental cattle, pre-weaned calves and male cattle, highlighting breed, age, and sex as key risk factors.] We have added the information in the Conclusion in Page 10, Paragraph 2-3, Lines 229-302.

4. Response to Comments on the Quality of English Language

Point 1: The English is fine and does not require any improvement.

Response 1: There is no content that requires a reply.

5. Additional clarifications

[There is no content that requires additional clarifications.]

Reviewer 3 Report

Comments and Suggestions for Authors

Occurrence and Molecular Characterization of Cryptosporidium  spp. in beef Cattle in Yunnan Province, China

The article examined the occurrence of Cryptosporidium spp. in beef cattle in Yunnan Province, China. we collected 735 fecal samples  from six breed of beef cattle in five regions of Yunnan with the PCR and DNA sequencing.

In the abstract the conclusion: Further control strategies should be implemented to ensure the food safety and reduce the eco-nomic losses in beef cattle farms. It's very general and not specific, that could be the conclusion of any work. Do another one.

All keywords must be different from the title.

The introduction should be expanded with greater importance in single health mainly and worldwide distribution of the organism.

The methodology is viable and well described.

In the results, the titles of the figures and tables should be more descriptive and this will make the figures and tables self-descriptive.

The discussion should include the importance of zoonotic diseases and the possible risks to humans, the unique health implications and possible control measures, as proposed in the conclusion.

Author Response

Response to Reviewer 3 Comments

1. Summary

Thank you very much for taking the time to review this manuscript. Please find the detailed responses below and the corresponding revisions in the re-submitted files.

2. Questions for General Evaluation

Reviewer’s Evaluation

Response and Revisions

Does the introduction provide sufficient background and include all relevant references?

Can be improved

Improved in Page 2, Lines 67-78

Are all the cited references relevant to the research?

Yes

Many thanks

Is the research design appropriate?

Yes

Many thanks

Are the methods adequately described?

Yes

Many thanks

Are the results clearly presented?

Can be improved

Improved in Page 5, Lines 156-169

Are the conclusions supported by the results?

Can be improved

Improved in Page 1, Lines 37-41

3. Point-by-point response to Comments and Suggestions for Authors

Comments 1: [Occurrence and Molecular Characterization of Cryptosporidium spp. in beef Cattle in Yunnan Province, China]

Response 1: Many thanks for your positive comment.

Comments 2: [The article examined the occurrence of Cryptosporidium spp. in beef cattle in Yunnan Province, China. we collected 735 fecal samples from six breed of beef cattle in five regions of Yunnan with the PCR and DNA sequencing]

Response 2: Many thanks for your positive comment.

Comments 3: [In the abstract the conclusion: Further control strategies should be implemented to ensure the food safety and reduce the eco-nomic losses in beef cattle farms. It's very general and not specific, that could be the conclusion of any work. Do another one. ]

Response 3: We thank the reviewer for this suggestion. The original conclusion has been replaced with a targeted statement: [“To effectively reduce the prevalence of Cryptosporidium spp. from beef cattle in Yunnan, the implementation of proper sanitation management, rigorous rodent control, and farmer education programmes is crucial. These integrated measures are critical for maintaining herd health, reducing economic losses, and ensuring meat safety across the province.”] This change can be found in Page 1, Lines 37-41.

Comments 4: [All keywords must be different from the title. ]

Response 4: Thank you for pointing this out. We have revised the keyword to: [Bovine, cryptosporidiosis, molecular epidemiology, risk factor, genotyping.] This change can be found in Page 1, Lines 42.

Comments 5: [The introduction should be expanded with greater importance in single health mainly and worldwide distribution of the organism. ]

Response 5: Thank you for your suggestion. We expanded the introduction as following: [Yunnan Province, located in southwestern China, provides a high-quality production environment for beef cattle production due to its suitable geographical and climatic conditions [19]. As Chinese largest beef cattle production province, it maintained over 9 million head of cattle in 2023 and is home to at least 15 distinct cattle breeds [19]. ……. The “One Health” framework emphasizes integrated approaches to reduce disease risks at the human-animal-environment interface. A better understanding of Cryptosporidium species in beef cattle and their transmission potential can lead to better control strategies. These measures would improve animal health and production efficiency, thus supporting the growing demand for sustainable food production.] This change can be found in Page 1, Lines 67-78.

Comments 6: [The methodology is viable and well described. ]

Response 6: Many thanks for your positive comment.

Comments 7: [In the results, the titles of the figures and tables should be more descriptive and this will make the figures and tables self-descriptive. ]

Response 7: Thank you for pointing this out. We have changed the titles of Table 1, Table 2 and figure 2, which can be found in Page 3, Lines 126, Page 6, Lines 202 and Page 8, Lines 208.

Comments 8: [The discussion should include the importance of zoonotic diseases and the possible risks to humans, the unique health implications and possible control measures, as proposed in the conclusion. ]

Response 8: Thank you for pointing this out. [“To meet the growing demand for meat products, Yunnan Province has significantly expanded the beef cattle industry. However, large numbers of animals in confined spaces could promote the transmission of Cryptosporidium spp.. Both farmers and consumers should be aware of cattle-associated Cryptosporidium species, as these pathogens can spread through fecal contamination of water, soil, and food, potentially exposing humans to infectious oocysts. This poses a particular risk to immunocompromised individuals, as there is no effective treatment against cryptosporidiosis. Our study identified key risk factors including overcrowding, poor sanitation and insufficient rodent control, all of which contribute to the transmission of Cryptosporidium spp. in beef cattle. Only through implementing the “One Health” approach, which emphasizes human-animal-environment interactions, can effectively mitigate Cryptosporidium infections [70].”] This change can be found in Page 10, Paragraph 2, Lines 280-290.

4. Response to Comments on the Quality of English Language

Point 1: The English is fine and does not require any improvement.

Response 1: There is no content that requires a reply.

5. Additional clarifications

[There is no content that requires additional clarifications.]

Round 2

Reviewer 1 Report

Comments and Suggestions for Authors

The authors took into consideration all the reviewer's suggestions and made the necessary corrections.